# Efficient and Stable Deep-Blue 0D Copper-Based Halide TEA_2_Cu_2_I_4_ with Near-Unity Photoluminescence Quantum Yield for Light-Emitting Diodes

**DOI:** 10.3390/nano14231919

**Published:** 2024-11-28

**Authors:** Fang Yuan, Xiaoyun Liu, Songting Zhang, Peichao Zhu, Fawad Ali, Chenjing Zhao, Shuaiqi He, Qianhao Ma, Jingrui Li, Kunping Guo, Lu Li, Zhaoxin Wu

**Affiliations:** 1Key Laboratory for Physical Electronics and Devices of the Ministry of Education & Shaanxi Key Lab of Information Photonic Technique, School of Electronic Science and Engineering, Xi’an Jiaotong University, Xi’an 710049, China; yuanf121@xjtu.edu.cn (F.Y.); fawad_ali@stu.xjtu.edu.cn (F.A.); shuaiqihe@stu.xjtu.edu.cn (S.H.);; 2Branch of LONGi Green Energy Technology Co., Ltd. in XixianNew Area, No. 215 Jinggan Second Street, Yongle Town, Jinghe Xincheng, Xixian New Area, Xi’an 710018, China; 3Electronic Materials Research Laboratory, Key Laboratory of the Ministry of Education, International Center for Dielectric Research and International Joint Laboratory for Micro/Nano Manufacturing and Measurement Technology, School of Electronic Science and Engineering, Xi’an Jiaotong University, Xi’an 710049, China; 4School of Electronic Information and Artificial Intelligence, Shaanxi University of Science and Technology, Xi’an 710021, China; 5Collaborative Innovation Center of Extreme Optics, Shanxi University, Taiyuan 030006, China

**Keywords:** 0D copper-based halides, deep-blue LEDs, high PLQY, stability

## Abstract

Achieving deep-blue light with high color saturation remains a critical challenge in the development of white light-emitting diode (LED) technology, necessitating luminescent materials that excel in efficiency, low toxicity, and stability. Here, we report the synthesis of [N(C_2_H_5_)_4_]_2_Cu_2_I_4_ (TEA_2_Cu_2_I_4_) single crystals (SCs), which exhibit deep-blue photoluminescence (PL) at 450 nm. These crystals are characterized by a significant Stokes shift of 180 nm, a long lifetime of 1.7 μs, and an impressive photoluminescence quantum yield (PLQY) of 96.7% for SCs and 87.2% for polycrystalline films. The zero-dimensional structure is attributed to the proper spacing of triangular inorganic units [Cu_2_I_4_]^2−^ by organic cations [N(C_2_H_5_)_4_]^+^. This structural arrangement facilitates broadband deep-blue light emission with phosphorescent characteristics, as evidenced by temperature-dependent PL and time-resolved photoluminescence (TRPL) measurements. The band gap properties of TEA_2_Cu_2_I_4_ were further elucidated through density functional theory (DFT) computations. Notably, the material exhibited minimal PL intensity degradation after continuous UV irradiation and one month of exposure to ambient conditions. Moreover, the polycrystalline film of TEA_2_Cu_2_I_4_ maintained substantial deep-blue emission even after one year of storage. Utilizing TEA_2_Cu_2_I_4_ thin film, we fabricated an electroluminescent device emitting deep-blue light with high color saturation, featuring CIE coordinates (0.143, 0.076) and a brightness of 90 cd/m^2^. The exceptional photophysical properties of TEA_2_Cu_2_I_4_ render it a highly promising candidate for optoelectronic applications.

## 1. Introduction

The development of stable and efficient primary color materials (blue, green, and red), particularly those emitting deep-blue light with high color saturation, indicating purity and intensity crucial for enhancing visual perception and minimizing power consumption, is pivotal for propelling full-color display technology forward [1]. While significant strides have been made in red- and green-light materials for both organic light-emitting diodes (OLEDs) and perovskite light-emitting diodes (PeLEDs), with external quantum efficiencies (EQEs) surpassing 20% and deemed adequate for commercialization [2,3,4,5], blue PeLEDs lag slightly behind their counterparts [6,7,8,9]. Achieving a deep-blue light-emitting material with high color saturation is imperative to meet the National Television Standards Committee (NTSC) standards [10,11]. This necessity arises from three primary factors: firstly, enhanced color saturation is crucial for improving the human eye’s visual perception, necessitating a higher saturation of blue light [12]; secondly, the CIE coordinate y value of blue light significantly influences the power consumption of full-color displays, with a lower CIE y value substantially reducing device power consumption [13]; and finally, the wide band gap of deep-blue light facilitates the generation of other colors through energy transfer, simplifying the manufacturing process of full-color displays and enhancing device stability [14]. Despite these advantages, creating a stable, deep-blue light-emitting material with high color saturation remains a challenge due to the limited building blocks available for blue-light materials with large band gaps [6,15,16].

Recently, copper-based halides have emerged as promising alternatives to lead-based halide perovskites in the realm of deep-blue light-emitting material with high color saturation, owing to their ultra-high photoluminescence quantum yield (PLQY), non-toxicity, and abundance [17,18,19,20,21,22,23,24,25]. Significant progress has been made with lead-based halide perovskite materials, such as the successful synthesis of ultra-thin (PEA)_2_PbBr_4_ nanosheets and (P-PDABr_2_)_2_PbBr_4_ nanocrystals by Deng et al. and Yuan et al. in 2019, achieving PLQYs of 25% and 77%, respectively [26,27]. These materials have been used in electroluminescent (EL) devices with CIE y-coordinate values of ≤0.08, aligning well with the deep-blue NTSC standard (0.14, 0.08). However, despite their high PLQY, producing dense nanocrystal films via spin-coating without a decrease in PLQY remains a challenge [28]. Additionally, the toxicity of lead (Pb) poses severe limitations on large-scale production [29]. In response to these challenges, the development of copper-based halides has garnered considerable attention. Among them, the all-inorganic halide Cs_3_Cu_2_I_5_ stands out, emitting a wide range of blue light with a large Stokes shift and a relatively high PLQY, making it an attractive candidate for optoelectronic applications [30,31,32,33,34,35]. Additionally, EL devices based on Cs_3_Cu_2_I_5_ nanocrystals have also been reported, with an EQE of up to 1.12% [36]. Furthermore, the halides Rb_2_CuX_3_ (X = Cl, Br) and K_2_CuX_3_ (X = Cl, Br), developed by D. Creason et al., demonstrate PLQYs exceeding 60% in the deep-blue light region (385–395 nm) [37,38]. The remarkable emission of these copper halides with low-dimensional crystal structures is attributed to self-trapped excitons (STEs) generated by a soft lattice and strong charge concentration [39]. However, despite these advantages, solution-processed films of such all-inorganic copper halide materials often suffer from severe aggregation-induced quenching, leading to a drastic reduction in the PLQY. Therefore, the development of lead-free, copper-based deep-blue light-emitting materials with high color saturation, high efficiency, and remarkable stability remains a critical research focus.

In this work, we report the successful synthesis of the zero-dimensional (0D) copper-based halide [N(C_2_H_5_)_4_]_2_Cu_2_I_4_ (TEA_2_Cu_2_I_4_) as a highly efficient and stable blue emitter, achieved through a straightforward approach. In the 0D structure, [Cu_2_I_4_]^2−^ dimers, consisting of two edge-sharing triangles, are effectively separated by tetraethylammonium cations [N(C_2_H_5_)_4_]^+^, resulting in a stable crystal structure. Compared to the bromine compounds we reported earlier [24], the substitution of Br with I in the copper-based halide structure results in TEA_2_Cu_2_I_4_ exhibiting a shorter emission wavelength (450 nm vs. 463 nm for TEA_2_Cu_2_Br_4_) and a larger band gap (3.73 eV vs. 2.68 eV for TEA_2_Cu_2_Br_4_), with an exceptional PLQY of 96.7%, representing the highest PLQY for deep-blue emission reported thus far. Consistent with other copper-based halides, its broadband blue emission exhibits a significant Stokes shift (180 nm) and a long PL lifetime of 1.7 μs. Temperature-dependent PL and time-resolved TRPL characterization further confirm that efficient blue light emission is attributed to phosphorescence. Furthermore, TEA_2_Cu_2_I_4_ exhibits exceptional hydrophobic and thermal stability, with a decomposition temperature reaching up to 260 °C and the ability to be stored in air for approximately 30 days with minimal loss in PL intensity. Notably, an EL device utilizing spin-coated thin films was effective, with CIE coordinates (0.143, 0.076) fully matching the NTSC standard for deep-blue light with high color saturation. TEA_2_Cu_2_I_4_ stands out as an exciting material for optoelectronic applications, thanks to its superior stability, efficient deep-blue light with high color saturation, and solution processing characteristics.

## 2. Results and Discussion

Figure 1 presents the structural characterization of the TEA_2_Cu_2_I_4_ SCs derived from single-crystal X-ray diffraction (SCXRD) data collected at 150 K. Detailed interatomic distances and angles are provided in Appendix A. High-quality, colorless TEA_2_Cu_2_I_4_ single crystals were synthesized by mixing tetraethylammonium iodide (TEAI) and copper(I) iodide (CuI) precursors in a 1:1 molar ratio using acetone as the solvent. The solution was filtered and allowed to stand (see the Appendix A for experimental details). Previous reports have noted the presence of [Cu_2_I_4_]^2−^ dimers in solution as anions, but without structural information on the SCs [40]. The TEA_2_Cu_2_I_4_ crystal adopts a linear structure belonging to the P21/c space group, with the lattice constants a = 8.57080 Å, b = 14.20100 Å, and c = 11.19830 Å, and the angles α = γ = 90° and β = 96.48°. Figure 1a displays the unit cell of TEA_2_Cu_2_I_4_, consisting of an inorganic anionic [Cu_2_I_4_]^2−^ dimer and a tetraethylammonium cation [N(C_2_H_5_)_4_]^+^. In the [Cu_2_I_4_]^2−^ dimer, two copper atoms occupy the centers of two edge-sharing triangles, and these isolated units are periodically separated by [N(C_2_H_5_)_4_]^+^, forming a 0D structure, as depicted in Figure 1b. We infer that the flexible [N(C_2_H_5_)_4_]^+^ cations embedded may cause lattice distortions, potentially aiding the formation of STEs in TEA_2_Cu_2_I_4_. However, the exact role of [N(C_2_H_5_)_4_]^+^ cations or [Cu_2_I_4_]^2−^ dimers in this process is uncertain. Further studies are needed to understand this mechanism. In the 0D structure of TEA_2_Cu_2_I_4_, the organic [N(C_2_H_5_)_4_]^+^ cations isolate the [Cu_2_I_4_]^2−^ units, providing a spacing effect that reduces aggregation-induced quenching. The tetraethylammonium cation induces an effect akin to quantum confinement throughout the structure, potentially contributing to the high PLQY of TEA_2_Cu_2_I_4_. A comparison of the XRD data of the synthesized TEA_2_Cu_2_I_4_ SCs with the theoretically calculated data reveals good overall agreement, as shown in Figure 1c. The simulated PXRD pattern was generated using the refined structural parameters from SCXRD analysis. Additionally, X-ray photoelectron spectroscopy (XPS) was employed to investigate the chemical composition and valence states of the synthesized SCs. The XPS results, illustrated in Appendix A, confirm the presence of C, N, I, and Cu elements. Notably, the Cu 2p spectrum exhibits two peaks at approximately 951.6 eV and 931.8 eV, corresponding to the 2p1/2 and 2p3/2 states of monovalent copper, respectively. The absence of a distinct signal at 943 eV, typically associated with Cu(II), indicates the absence of divalent copper in TEA_2_Cu_2_I_4_. The high-resolution I 3d spectrum displays two peaks separated by 11.5 eV at 629.6 eV and 618.1 eV, corresponding to the I 3d3/2 and 3d5/2 states, respectively, as shown in Appendix A. These findings corroborate the successful synthesis of a high-quality SC using a straightforward method. Furthermore, a polycrystalline film of TEA_2_Cu_2_I_4_ was prepared by dissolving the precursors (TEAI:CuI = 1:1) in dimethyl sulfoxide (DMSO) overnight and spin-coating the solution in a single step.

The optical characteristics of the as-prepared TEA_2_Cu_2_I_4_ thin film were investigated, and the results are presented in Figure 2. Specifically, Figure 2a displays a PL spectrum featuring a broad blue emission centered at 450 nm, with a full-width-at-half-maximum (FWHM) of 72 nm under 270 nm excitation. The observation of a single peak indicates the presence of a sole radiative channel. As illustrated in Figure 1d, upon exposure to UV radiation at 254 nm, the TEA_2_Cu_2_I_4_ SC emits a vivid, deep-blue light. Notably, the PLQY was measured to be exceptionally high, reaching 87.2% for thin films and 96.7% for SCs at room temperature. It is worth highlighting that the PLQY of TEA_2_Cu_2_I_4_ ranks among the highest reported efficiencies, a remarkable feat considering the typical aggregation-induced quenching effect observed in spin-coated fluorescent films [41]. The optical absorption spectra reveal a distinct peak at 276 nm, corresponding to a band gap of 3.73 eV, calculated using the Tauc equation, with minimal self-absorption observed. Furthermore, time-resolved photoluminescence (TRPL) decay measurements conducted at room temperature exhibited a decay curve consistent with a monoexponential model, yielding a lifetime of 1.7 μs, which is comparable to other copper-based halides [42,43]. As depicted in Figure 2c, the PL emission peaks, spanning from 240 nm to 340 nm excitation, consistently occurred at 450 nm, suggesting that exciton relaxation initiates from the same excited state. This, coupled with the TRPL findings, confirms a single-channel radiation mechanism. Notably, the maximum PL emission is observed at an excitation wavelength of 270 nm, accompanied by a significant Stokes shift of 180 nm, further indicating reduced self-absorption. These results collectively demonstrate that the luminescence mechanism of TEA_2_Cu_2_I_4_ is not attributed to multichannel radiation. Instead, the combination of strong PLQY, low self-absorption, and extended lifetime underscores the exceptional photophysical properties of TEA_2_Cu_2_I_4_ as a luminescent material, highlighting its potential for various optoelectronic applications.

To gain a deeper insight into the broadband deep-blue emission process, temperature-dependent PL spectra were meticulously analyzed, as depicted in Figure 3a–d. At 80 K, the PL peak exhibits a narrow FWHM, attributable to reduced atomic vibrations and the stabilization of the localized electronic excited state at low temperatures. As the temperature increases, the electron–phonon coupling strengthens, leading to a progressive broadening of the FWHM. It is noteworthy that the PL peak position remains virtually unchanged, aligning with the characteristics of luminescence originating from the triplet state, as evident in Figure 3b [44]. The strength of electron–phonon coupling can also be characterized by calculating the Huang–Rhys factor S using the following equation:$$\mathrm{FWHM}\left(T\right)=2.36\sqrt{S}E_{\mathrm{ph}}{\left[\coth\left(\frac{E_{\mathrm{ph}}}{{2k}_{B}T}\right)\right]}^{1/2}$$
where *T* is the temperature, *E*_ph_ is the effective photon energy, *S* is the Huang–Rhys factor, and *k*_B_ is the Boltzmann constant. By fitting the FWHM and temperature data, the *S* factor and phonon frequency were determined to be 40 and 13.38 meV, respectively. The significantly larger *S* factor compared to other light-emitting materials, such as CdSe and ZnSe [45,46], indicates a strong electron–phonon coupling interaction, which is crucial for the observed broadband emission. Furthermore, we measured the integrated PL intensity at various temperatures, revealing only slight variations (Figure 3c). To explore the blue emission mechanism more deeply, temperature-dependent TRPL measurements were conducted (Figure 3e). The decay curves, recorded over a temperature range of 30 to 300 K, were well fitted with a single exponential function (Figure 3e). As illustrated in Figure 3f, TEA_2_Cu_2_I_4_ exhibits a negligible temperature dependence with an average lifespan of 1.8 μs, deviating from the typical temperature-dependent PL decay durations observed in fluorescence. Coupled with the temperature-dependent PL spectra, these properties suggest phosphorescence as the underlying mechanism. The deep-blue emission of TEA_2_Cu_2_I_4_ can be attributed to phosphorescence, occurring via the radiative recombination of electrons in the lowest excited triplet state (T_1_) with holes in the ground state (S_0_) after intersystem crossing (ISC), as depicted in Appendix A. The substantial optical band difference (ΔE) between the S1 and T1 states results in a significant Stokes shift. Figure 3g displays Raman spectra at 100 and 300 K, excited with a 532 nm wavelength. Decreasing the temperature to 80 K restricts phonon vibrations, leading to a notable decrease in intensity in the 0–150 cm^−1^ region. Conversely, as the temperature rises to 300 K, the intensity of phonon vibrations gradually increases, further confirming the strong electron–phonon coupling in TEA_2_Cu_2_I_4_. To exclude the possibility that the emission mechanism is due to persistent defects, power-dependent PL measurements were performed (Figure 3h). The PL intensity exhibits a linear dependence on excitation power, indicating the absence of emission defects at varying excitation intensities. In addition, the PL of the as-prepared TEA_2_Cu_2_I_4_ single crystal and its powder after grinding was compared in Appendix A, and it was found that the PL peak position did not change, and the PL intensity decreased sharply after grinding, ruling out the possibility that the luminescence of TEA_2_Cu_2_I_4_ is caused by surface defects. Based on these findings, the deep-blue luminescence process was attributed to phosphorescence arising from triplet state relaxation, highlighting the unique optical properties of TEA_2_Cu_2_I_4_.

To gain a deeper understanding of the PL synthesis in TEA_2_Cu_2_I_4_, we conducted a comprehensive study combining theoretical and experimental analyses. Utilizing the Perdew–Burke–Ernzerhof exchange correlation functional for solids (PBEsol) within the framework of first-principles density functional theory (DFT) [47,48,49], we employed the FHI-AIMS code, which features an all-electron numeric atom-centered orbital method. Our DFT calculations not only provide insights into the band structure but also highlight that, although the organic [N(C_2_H_5_)_4_]^+^ cations do not directly contribute to the charge transfer processes due to the localization of charges primarily on the inorganic [Cu_2_I_4_]^2−^ dimers, there is a spatial separation facilitated by these organic cations. In the 0D structure of TEA_2_Cu_2_I_4_, the [N(C_2_H_5_)_4_]^+^ cations act as effective spacers between the [Cu_2_I_4_]^2−^ dimers, reducing aggregation-induced quenching. The structural arrangement with the organic cations plays a crucial role in maintaining the stability and optical properties of TEA_2_Cu_2_I_4_. The band structure analysis (Figure 4a) reveals that the conduction band minimum (CBM) is primarily composed of Cu 4s orbitals. These orbitals, although localized within each [Cu_2_I_4_]^2−^ dimer, exhibit sufficient spatial delocalization to allow for some charge transport between neighboring [Cu_2_I_4_]^2−^ dimers. The bending of the energy bands near the CBM indicates that there is charge dispersion across the structure, enabling limited charge mobility despite the 0D nature of TEA_2_Cu_2_I_4_. The organic [N(C_2_H_5_)_4_]^+^ cations primarily serve as spacers between the [Cu_2_I_4_]^2−^ dimers, regulating their distance to minimize aggregation-induced quenching and maintain the optical properties. These cations do not directly participate in charge transfer but rather facilitate the stable isolation of the [Cu_2_I_4_]^2−^ dimers while allowing for limited charge mobility. The band structure (Figure 4a) indicates that the valence band (VB) consists of a succession of flat bands with minimal dispersion. Consequently, both the projected direct band gap (3.76 eV) and the indirect band gap (3.75 eV) exhibit near-equivalence. Although incorporating precise exchange functions could be a potential avenue, it comes at a substantial computational cost and was omitted from our analysis due to its marginal influence on the energetics of the valence and conduction bands. Notably, the conduction band (CB) edge demonstrates significant dispersion, suggesting poor hole transport but favorable electron transport properties, based on the disparity in dispersion between the VB and CB edges. For clarity, we denote the energy bands successively lower than the VB maximum (VBM) as VBM-1, VBM-2, and so forth. Figure 4b illustrates the wave functions spanning from VBM to VBM-4 at the Γ point. These VB states are primarily composed of Cu 3d and I 4p closed-shell atomic orbitals, with minimal contributions from organic components. Within each primitive cell, comprising two TEA_2_Cu_2_I_4_ units, these VB wave functions can be expressed as linear combinations of contributions from the two chemically identical [Cu_2_I_4_]^2−^ anions. Specifically, VBM-1 and its counterpart differ only in the sign of their linear combination, as do VBM-2, VBM-3, and VBM-4. The localized nature of closed-shell atomic orbitals, particularly Cu 3d, results in the high localization of these VB states at each [Cu_2_I_4_]^2−^ anion. This localization leads to two critical outcomes: (i) the formation of flat bands due to weak interactions between adjacent [Cu_2_I_4_]^2−^ anions, and (ii) the near-degeneracy observed between paired states, such as VBM and VBM-1, and VBM-2 and VBM-3. The wave function of the conduction band minimum (CBM) exhibits unique characteristics. It is primarily formed by the 4s orbitals of both Cu atoms within each [Cu_2_I_4_]^2−^ anion, which are far more spatially delocalized than the Cu 3d orbitals found in the VB states. This delocalization extends across individual [Cu_2_I_4_]^2−^ anions and is further coupled by I atoms and TEA^+^ cations, resulting in significant band dispersion. In summary, the DFT-calculated band structure highlights multiple potential electrical transition pathways, including transitions between various VBs, such as VBM and VBM-2, and the CBM, as well as the reverse transitions. These findings underscore the rich electronic landscape of TEA_2_Cu_2_I_4_, offering insights into its PL properties.

Based on the above theoretical and experimental analyses, a plausible PL mechanism for TEA_2_Cu_2_I_4_ is proposed as follows: (i) Multiple channels for electronic excitation may exist, corresponding to the transition of a single electron from different VB states to the CBM band. Specifically, the transition from the VBM or its nearby states to the CBM corresponds to the excitation from the ground state (S_0_) to the first singlet excited state (S_1_), with an absorption energy of 3.73 eV. Similarly, transitions from deeper VB states such as VBM-2 or VBM-3 to the CBM are associated with excitation to higher singlet states like S_2_. (ii) Following excitation to the singlet states S_1_ and S_2_, the systems may undergo an ultrafast ISC process, transitioning to the corresponding triplet state T_1_. The primary difference between the singlet state S_1_ and the triplet state T_1_ lies in the localized hole wave functions. Consequently, the ISC process results in a similar magnitude of energy-level lowering for both transitions. (iii) Subsequently, radiative recombination from the triplet state T_1_ back to the ground state S_0_ may then emit phosphorescence at an energy of 2.76 eV (Appendix A). However, it should be noted that this is one of several possible interpretations, and further experimental and theoretical investigations are required to conclusively determine the exact PL mechanism of TEA_2_Cu_2_I_4_.

Figure 5 presents the structural and PL stability of TEA_2_Cu_2_I_4_. As depicted in Figure 5a, the XRD diffraction peaks of TEA_2_Cu_2_I_4_ powders remained virtually unaltered, with no emergence of new phase peaks even after three months in an ambient environment. This observation underscores the significant role played by the triangular-shaped [Cu_2_I_4_]^2−^ units, formed by monovalent copper atoms, in ensuring the structural stability of the material. Additionally, thermogravimetric analysis reveals a decomposition temperature of approximately 220 °C for TEA_2_Cu_2_I_4_, indicating thermal stability on par with pure inorganic materials [50]. The PL stability of TEA_2_Cu_2_I_4_ was also assessed, as illustrated in Figure 5c–e. After being exposed to an open-air environment for one month, the PL intensity exhibited a moderate decrease, yet the peak position remained practically unchanged. Notably, the PLQY decreased by about 10%, as evidenced in Figure 5d. After storing the prepared TEA_2_Cu_2_I_4_ polycrystalline thin film in an atmospheric environment for one year, we observed a significant attenuation in its fluorescence. However, as depicted in Appendix A, the film still emitted a relatively bright blue light even after a year of storage. This experimental result underscores the remarkable stability of TEA_2_Cu_2_I_4_ in the atmospheric environment. These findings demonstrate that TEA_2_Cu_2_I_4_ exhibits resistance to oxidation and can be stored under atmospheric conditions. To further gauge its stability for practical applications, a continuous exposure to a 254 nm UV lamp was conducted. As depicted in the contour diagram in Figure 5e, the PL intensity of TEA_2_Cu_2_I_4_ powder initially increased over the first ten hours before stabilizing, with the peak position maintaining consistency throughout. In summary, the remarkable structural integrity and stability of TEA_2_Cu_2_I_4_, even in atmospheric conditions at room temperature, highlight its potential as a luminous material for real-world applications. These findings underscore the material’s robustness and suggest promising avenues for future research and development.

In addition to exhibiting a high PLQY, TEA_2_Cu_2_I_4_ boasts a substantial Stokes shift, rendering it highly suitable as an emitter in EL devices. To demonstrate its potential in light-emitting diodes, we fabricated an EL device utilizing a spin-coated TEA_2_Cu_2_I_4_ thin film, as depicted in Figure 6a. The performance of the TEA_2_Cu_2_I_4_-based blue LEDs, constructed with a Glass/ITO/PEDOT:PSS/TEA_2_Cu_2_I_4_/1,3,5-tris(1-phenyl-1H-benzimidazol-2-yl) benzene (TPBi)/LiF/Al structure, was meticulously analyzed. The thicknesses of the PEDOT:PSS, TEA_2_Cu_2_I_4_, TPBi, LiF, and Al layers were measured to be 45, 30, 55, 1, and 100 nm, respectively, using a step profiler. To verify the device structure’s integrity, UV photoelectron spectroscopy (UPS) measurements were conducted on the TEA_2_Cu_2_I_4_ layer, as illustrated in Figure 6b. During data processing, the reference gold’s work function was adjusted to 5.1 eV. The results indicated that TEA_2_Cu_2_I_4_ possesses a Fermi energy level of approximately 4.25 eV (Figure 6b, left panel). By subtracting the excitation energy (He I excitation, 21.22 eV) from the intercept at 18.59 eV (cut-off region, Figure 6b, right panel), the valence band maximum (VBM) of TEA_2_Cu_2_I_4_ was determined to be around 6.88 eV. The conduction band minimum (CBM) was calculated to be approximately 3.15 eV, based on the compound’s energy band gap of 3.73 eV, as shown in the Tauc plot (Figure 2a). For the device to operate, the ITO layer was subjected to a forward voltage, and the corresponding brightness spectrum was captured from the glass’s backside. Figure 6c displays the current density and luminance curves of the LED at various voltages. Owing to the voltage drop across the interface barrier, the diode exhibits a conventional characteristic curve and a relatively high turn-on voltage of ~3.8 V [51]. Both current density and brightness escalate rapidly with increasing voltage, reaching a maximum at 6.8 V with a luminance of 90 cd/m^2^. An operational photograph of the deep-blue LED is provided in the inset of Figure 6c. The LED’s EQE is about 0.45%, which still has a lot of room for improvement in the future, mainly due to energy level potential barriers within the device that result in an imbalance of electron–hole injection (Figure 6d). As a next step, we aim to optimize the device structure to enhance its performance further. The TEA_2_Cu_2_I_4_-based LED devices also show a high reproducibility (Appendix A), which originates from the superior uniformity and stability of TEA_2_Cu_2_I_4_. Additionally, Appendix A showcases a comparative analysis of the PL and EL spectra, alongside an evaluation of the stability under varying voltages. This comprehensive assessment highlights the exceptional color stability of TEA_2_Cu_2_I_4_-based LEDs across a wide range of operating voltages. The EL spectra of the device align with the Commission Internationale de l’Eclairage (CIE) color coordinates (0.143, 0.076), as depicted in Figure 6e. Notably, the CIEy coordinate value of <0.08 meets the criterion for deep-blue light. As evidenced in Figure 6e, the device’s EL spectra correspond to CIE color coordinates of 0.143 and 0.076, fulfilling the requirement for deep-blue light with a CIEy coordinate value below 0.08. In future endeavors, we will focus on refining the device’s structure to elevate its functionality.

In comparing our current work on TEA_2_Cu_2_I_4_ with our previous work on TEA_2_Cu_2_Br_4_ [24], it is evident that the iodinated analog exhibits distinct optical and electronic properties that set it apart from its bromide counterpart. TEA_2_Cu_2_I_4_ demonstrates a shorter emission wavelength, larger band gap, and high PLQY, making it more suitable for high-saturation deep-blue light emission applications. Furthermore, the TEA_2_Cu_2_I_4_-based LED exhibits improved device performance with better EQE compared to the TEA_2_Cu_2_Br_4_-based device. These findings underscore the unique advantages of TEA_2_Cu_2_I_4_ and highlight the significance of our current work in expanding the understanding and application potential of copper-based halide materials.

## 3. Conclusions

In summary, we have achieved a significant milestone by successfully synthesizing high-quality single crystals of TEA_2_Cu_2_I_4_, marking the first instance of this accomplishment. Notably, the quantum confinement effect within the tetraethylammonium cations leads to an exceptional PLQY of 96.7% in the 0D structure. Moreover, solution-based thin films exhibit a remarkably high PLQY of 87.2%. Our DFT analyses, coupled with fitting results utilizing the Huang–Rhys factor, reveal a pronounced electron–phonon interaction. Temperature-dependent PL and TRPL experiments further corroborate that the observed deep-blue emission originates from phosphorescence, with a maximum brightness of 90 cd/m^2^ attained under forward bias. Impressively, the CIE coordinate’s y value of 0.076 underscores high color saturation in the deep-blue range. Collectively, our findings underscore the potential of TEA_2_Cu_2_I_4_ as a promising lead-free, low-toxicity luminous material, holding intriguing prospects for future applications in the photovoltaic domain.

## Figures and Tables

**Figure 1 nanomaterials-14-01919-f001:**
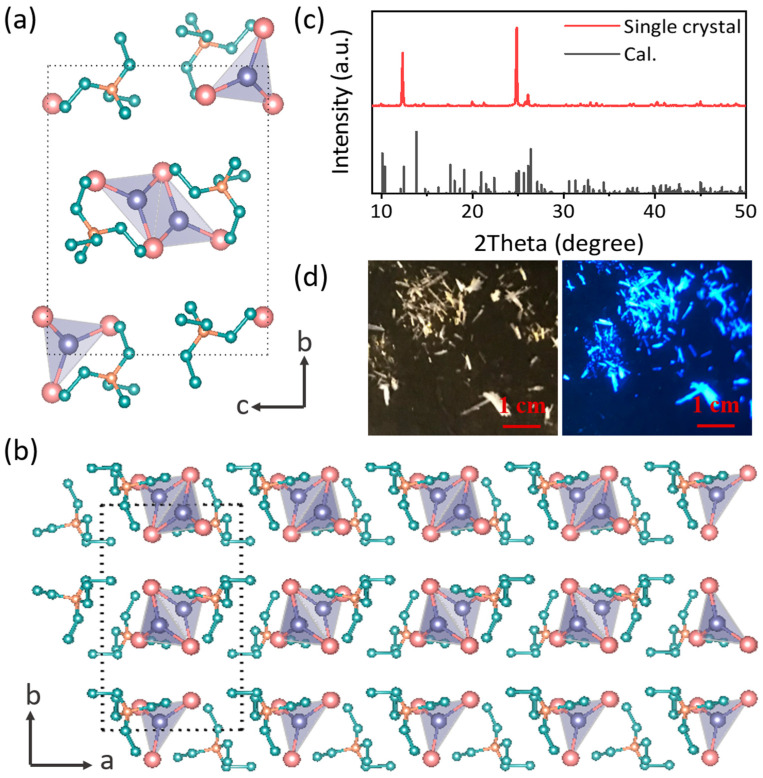
(**a**) The schematic view of the basic unit of TEA_2_Cu_2_I_4_ along the *a*-axis; (**b**) a polyhedral view of the structure of TEA_2_Cu_2_I_4_ projected along the *a*-axis; (**c**) the comparison of the experimental XRD data collected from a single crystal (red) and the simulated PXRD data (black) of TEA_2_Cu_2_I_4_ based on the refined structural model. (**d**) Photographs of the as-synthesized single crystal in daylight and under a 254 nm UV lamp; the scale bar is 1 cm.

**Figure 2 nanomaterials-14-01919-f002:**
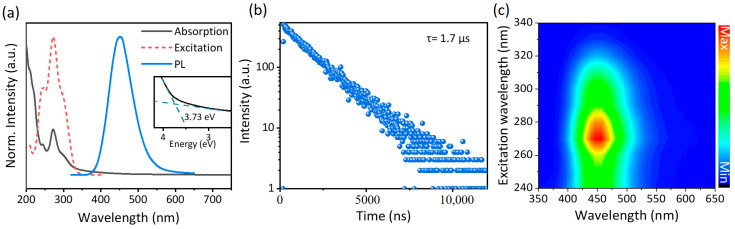
Optical properties of the TEA_2_Cu_2_I_4_ polycrystalline film. (**a**) Normalized TEA_2_Cu_2_I_4_ thin film steady-state absorption (black) and photoluminescence spectra (blue, λ_exc_ = 270 nm) (inset shows the determination of optical band gap using the Tauc equation). (**b**) Time-resolved PL decay curve of TEA_2_Cu_2_I_4_ at room temperature. (**c**) PL contour map of TEA_2_Cu_2_I_4_ thin film measured at room temperature under different excitation wavelengths.

**Figure 3 nanomaterials-14-01919-f003:**
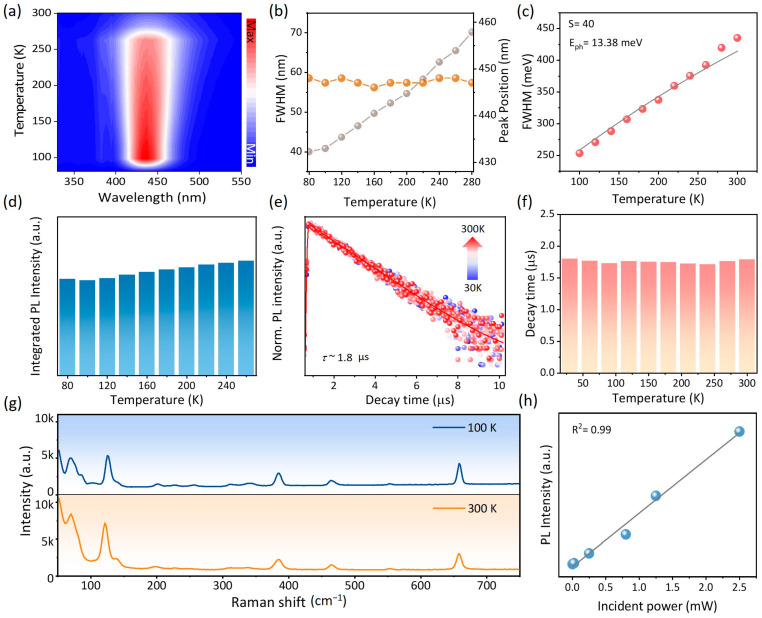
(**a**) Contour map of the temperature-dependent PL spectra; (**b**) corresponding FWHM and peak position of the PL spectra versus temperature; (**c**) fitting results of the FWHM (meV) as a function of temperature to obtain the Huang–Rhys factor S; (**d**) histogram of the variation in integrated PL intensity for the TEA_2_Cu_2_I_4_ thin film at different temperatures; (**e**) temperature-dependent time-resolved PL decay curve of TEA_2_Cu_2_I_4_ from 30 K to 320 K; (**f**) histogram of decay time with different temperatures; (**g**) Raman spectra of the TEA_2_Cu_2_I_4_ thin film at 100 K and 300 K. (**h**) PL intensity of TEA_2_Cu_2_I_4_ thin film under various incident light power levels.

**Figure 4 nanomaterials-14-01919-f004:**
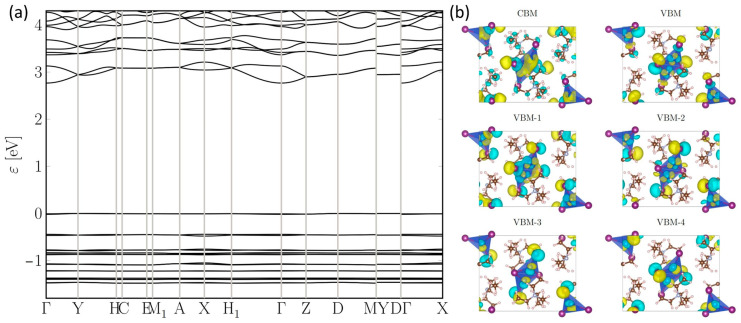
Electronic structure calculated with PBEsol density functional. (**a**) Band structure (with VBM shifted to 0 eV). (**b**) One-electron wave functions (with cyan and yellow presenting different signs) of VBM (VBM-1), (VBM-2), (VBM-3), (VBM-4) and CBM at Γ.

**Figure 5 nanomaterials-14-01919-f005:**
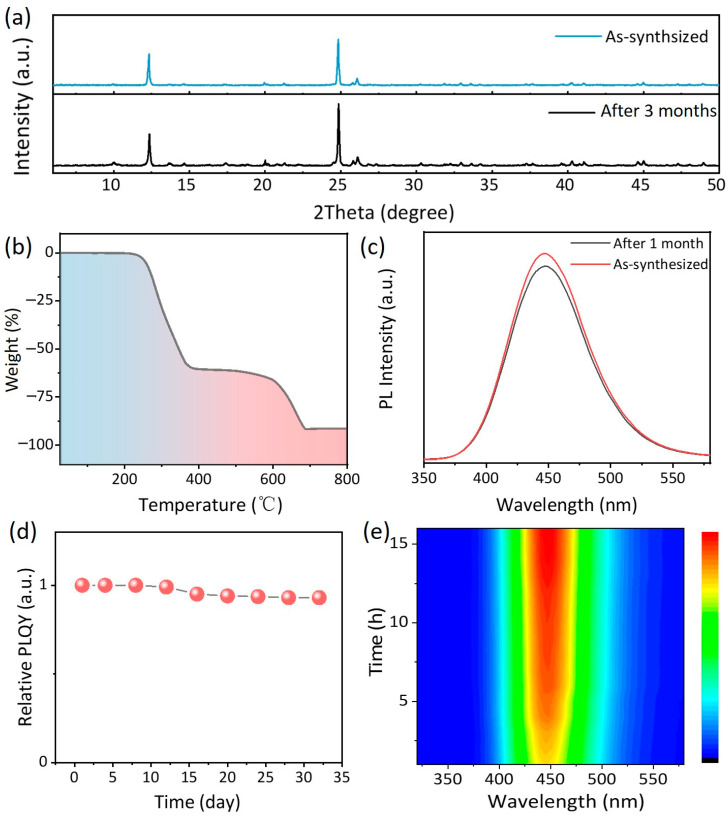
Stability of TEA_2_Cu_2_I_4_ is probed. (**a**) XRD pattern of the as-prepared TEA_2_Cu_2_I_4_ powder and after three months stored in the atmosphere; (**b**) thermogravimetric curve of TEA_2_Cu_2_I_4_ powder; (**c**) PL spectra of the as-prepared TEA_2_Cu_2_I_4_ and after 1 month; (**d**) the change in PLQY of TEA_2_Cu_2_I_4_ after one month in the atmosphere; (**e**) the contour plot of PL intensity of TEA_2_Cu_2_I_4_ under continuous 254 nm UV lamp irradiation.

**Figure 6 nanomaterials-14-01919-f006:**
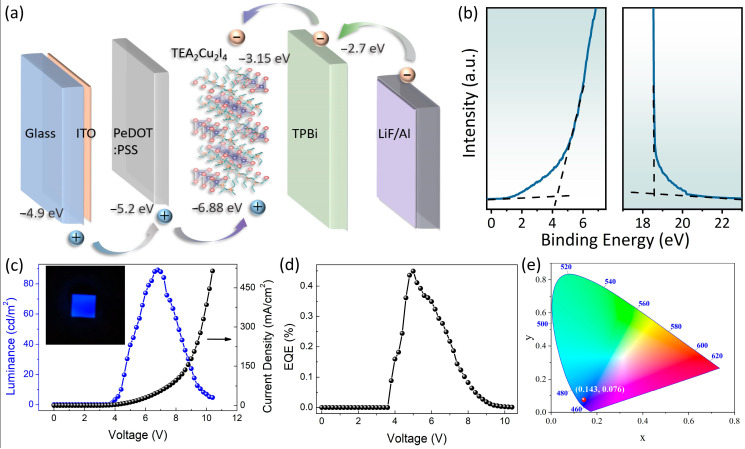
(**a**) Device structure of the LED; (**b**) UPS spectra of the TEA_2_Cu_2_I_4_ thin film; (**c**,**d**) luminance–voltage (*L–V*), current density–voltage (*J–V*), and EQE–voltage (*EQE–V*) curves of the device; (**e**) CIE coordinates corresponding to deep-blue LED device (red ball).

## Data Availability

Data are contained within the article and Appendix A.

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
