# Peer review of "Efficient and Stable Deep-Blue 0D Copper-Based Halide TEA2Cu2I4 with Near-Unity Photoluminescence Quantum Yield for Light-Emitting Diodes"

_nanomaterials, 2024, doi:10.3390/nano14231919_

Round 1

Reviewer 1 Report

Comments and Suggestions for Authors

Manuscript Number: nanomaterials-3283310

Comments

In this manuscript, Yuan et al described the synthesis of TEA2Cu2I4 single crystals and characterized them using different analytical tools such as single crystal XRD, PL, TRPL, TGA etc. Authors claim that they have achieved significant milestones after exploring TEA2Cu2I4 for deep-blue light-emitting diodes (LEDs) demonstrating ~0.4 EQE. As explained by the authors in the introduction, deep-blue emitting materials are important for LEDs and current work provides a viable, non-toxic, stable, and thin-film processable alternative material. However, the current manuscript is inconsistent and contains some overclaims without the support of experimental evidence. I offer the following important comments, that may help authors to further improve the current version. Therefore, I recommend reconsideration of the manuscript for major revision before its possible publication in nanomaterials.  

1.      The title does not provide clarity. What is ‘near unity’ about?

2.      To what exactly ‘saturated’ referred in the manuscript is unclear. In chemistry, this has different meanings. Therefore, its use at different places in the manuscript causes a lot of confusion.

3.      The second para of the introduction section is inconsistent, and comparison with lead halides is an unnecessary attempt.

4.      Can authors explain what are self-trapped excitons in this case? How do soft lattice and strong charge concentration promote self-trapping? What are the energy levels involved in self-trapping? Is there any role of Cu-I bonding? Whose triplet state is involved in the PL mechanism? The current experimental results are insufficient to conclude the PL mechanism.

5.      It has been claimed that organic chains separate inorganic dimers. What are these organic chains? Are those different than A-site utilized to form TEA2Cu2I4? Does single crystal XRD support their presence?

6.      Diffraction peaks around 9 and 29 degrees are not matching with the reference pattern. What are these secondary phases? Also, it contradicts the authors claim of excellent agreement between reference and experimental pattern. How experimental and PXRD are different (caption of Figure 1)? How reference XRD pattern is obtained?

7.      Page 3: There is inconsistency in naming the A-site cation. What is tetraethyl organic chain? and how these are different than the tetraethylammonium cation?

8.      Figure 2 caption misses inset description. Which Taucs equation used? What type of band gap (direct/indirect) authors have considered to get Taucs plot?

9.      Why Figure 2c shows line broadening on excitation between 275-280 nm? Why compound behaves differently specifically in this excitation range?

10.   Description about PL mechanism is ambiguous and described without convincing experimental support. Can authors provide explanation for their claim of ‘single-channel radiation mechanism’? Further explanation suggests emission from triplet states. This makes manuscripts inconsistent and scientifically weak. The data discussed in manuscript does not provide enough support to the PL mechanism provided in the supporting information. The S factor is compared with CsPbBr3, which is known for excitonic emission. This comparison is unnecessary and irrational.

11.   Figure 5 and respective captions are misleading, inconsistent, and promoting incredulity.

12.   TEA2Cu2I4 possess 0D structure. How charge transfer is possible between inorganic units without interconnection? How many devices were fabricated to conclude EQE? Authors should list at least 5 best device data in SI for further clarity. The EQE-voltage is ambiguous. Provide EL and PL comparison with stability over a range of voltage change. 

Comments on the Quality of English Language

None.

Author Response

The following is a point-by-point response to the reviewers' comments (The original comments are in black, and our responses are in blue).

Reviewers' comments:

Reviewer 1#: 

Comments:

In this manuscript, Yuan et al described the synthesis of TEA2Cu2I4 single crystals and characterized them using different analytical tools such as single crystal XRD, PL, TRPL, TGA etc. Authors claim that they have achieved significant milestones after exploring TEA2Cu2I4 for deep-blue light-emitting diodes (LEDs) demonstrating ~0.4 EQE. As explained by the authors in the introduction, deep-blue emitting materials are important for LEDs and current work provides a viable, non-toxic, stable, and thin-film processable alternative material. However, the current manuscript is inconsistent and contains some overclaims without the support of experimental evidence. I offer the following important comments, that may help authors to further improve the current version. Therefore, I recommend reconsideration of the manuscript for major revision before its possible publication in nanomaterials:

  1. The title does not provide clarity. What is ‘near unity’ about?

Response: Thank you for bringing this important issue to our attention. We appreciate your thorough evaluation and constructive feedback. You mentioned that the title does not provide clarity, specifically questioning what is meant by 'near-unity'. We apologize for any confusion this may have caused. In our title, "Efficient and Stable Deep-Blue Near-Unity 0D Copper-based Halides TEA2Cu2I4 for Light-Emitting Diodes" the term 'near-unity' refers to the high photoluminescence quantum yield (PLQY) of the copper-based halide TEA2Cu2I4. The PLQY of the material reaches 96.7% for single crystals, which is exceptionally close to 100% or 'unity'. To make the title more precise and understandable, we have modified it as follows: "Efficient and Stable Deep-Blue 0D Copper-based Halide TEA2Cu2I4 with Near-Unity Photoluminescence Quantum Yield for Light-Emitting Diodes". This revised title more clearly conveys the key attribute of TEA2Cu2I4, namely its high PLQY approaching 100%, and its application in light-emitting diodes. We also apologize sincerely for any inconvenience this may have caused in reading. Thank you again for your valuable feedback. We hope this clarification addresses your concern.

  1. To what exactly ‘saturated’ referred in the manuscript is unclear. In chemistry, this has different meanings. Therefore, its use at different places in the manuscript causes a lot of confusion.

Response: Thank you for your constructive feedback on our manuscript. We understand your concern regarding the use of the term 'saturated' and its potential for confusion due to its multiple meanings in chemistry. We apologize for any uncertainty this may have caused. In our manuscript, the term 'saturated' is used in the context of light emission to describe the purity and intensity of the deep-blue color produced by the copper-based halide TEA2Cu2I4. Specifically, 'saturated deep-blue light' refers to a light emission that meets the color standards for deep blue as defined by the National Television Standards Committee (NTSC), characterized by high color purity and intensity. To clarify the usage of 'saturated' and prevent confusion, we have replaced instances of 'saturated deep-blue light' with 'deep-blue light with high color saturation' in the following revisions throughout the manuscript (Page 1, 2, and 3). We also replaced instances of 'Utilizing TEA2Cu2I4 thin film, we fabricated a saturated deep-blue electroluminescent device with CIE coordinates (0.143, 0.076) and a brightness of 90 cd/m2.' with 'Utilizing TEA2Cu2I4 thin film, we fabricated an electroluminescent device emitting deep-blue light with high color saturation, featuring CIE coordinates (0.143, 0.076) and a brightness of 90 cd/m2.' (Page 1), instances of 'deep-blue emitters' with 'those emitting deep-blue light with high color saturation' (Page 1), instances of 'saturated deep-blue light-emitting material' with 'deep-blue light-emitting material with high color saturation' (Page 2), instances of 'As such, the development of lead-free, copper-based saturated deep-blue light materials with high efficiency and remarkable stability remains a critical research focus.' with 'Therefore, the development of lead-free, copper-based deep-blue light-emitting material with high color saturation, high efficiency, and remarkable stability remains a critical research focus.' (Page 2) in the following revisions throughout the manuscript.

Additionally, we add a brief explanatory sentence in the Introduction to define 'high color saturation' in the context of light emission as 'indicating purity and intensity crucial for enhancing visual perception and minimiz-ing power consumption' (Page 1) in the following revisions throughout the manuscript.

By making these revisions, we aim to eliminate any confusion regarding the term 'saturated' and ensure that the intended meaning is clear to readers from various backgrounds. Thank you again for bringing this issue to our attention. We hope these clarifications address your concern and improve the manuscript.

  1. The second para of the introduction section is inconsistent, and comparison with lead halides is an unnecessary attempt.

Response: We fully appreciate your observation that the second paragraph may have appeared inconsistent, especially given the focus of our work on lead-free alternatives. However, we believe it is important to mention lead-based halides perovskite in the context of establishing the broader landscape of luminescent materials for light-emitting diodes (LEDs). This allows us to highlight the unique advantages of our copper-based halide, TEA2Cu2I4, in terms of being lead-free and non-toxic.

We have revised the second paragraph to provide a more concise overview of the current state of luminescent materials for LEDs, particularly focusing on the advancements in lead-based halides perovskites and the need for lead-free alternatives. We have maintained the comparison with lead-based materials but have reframed it to emphasize the necessity for exploring lead-free options due to concerns about toxicity and sustainability. The revised second paragraph are as follows:

"Recently, copper-based halides have emerged as promising alternatives to lead-based halide perovskites in the realm of deep-blue light-emitting material with high color saturation, owing to their ultra-high photoluminescence quantum yield (PLQY), non-toxicity, and abundance [18-26]. Significant progress has been made with lead-based halide perovskites materials, such as the successful synthesis of ultra-thin (PEA)2PbBr4 nanosheets and (P-PDABr2)2PbBr4 nanocrystals by Deng et al. and Yuan et al. in 2019, achieving PLQYs of 25% and 77% respectively [27, 28]. These materials have been used in electroluminescent (EL) devices with CIE y-coordinate values of ≤0.08, aligning well with the deep-blue NTSC standard (0.14, 0.08). However, despite their high PLQY, producing dense nanocrystal films via spin-coating without a decrease in PLQY remains a challenge [29]. Additionally, the toxicity of lead (Pb) poses severe limitations on large-scale production [30]. In response to these challenges, the development of copper-based halides has garnered considerable attention. Among them, the all-inorganic halide Cs3Cu2I5 stands out, emitting a wide range of blue light with a large Stokes shift and a relatively high PLQY, making it an attractive candidate for optoelectronic applications [31-36]. Additionally, EL devices based on Cs3Cu2I5 nanocrystals have also been reported, with an EQE of up to 1.12% [37]. Furthermore, the halides Rb2CuX3 (X= Cl, Br) and K2CuX3 (X= Cl, Br), developed by D. Creason et al., demon-strate PLQYs exceeding 60% in the deep-blue light region (385–395 nm) [38, 39]. The remarkable emission of these copper halides with low-dimensional crystal structures is attributed to self-trapped excitons (STEs) generated by a soft lattice and strong charge concentration [40]. However, despite these advantages, solution-processed films of such all-inorganic copper halide materials often suffer from severe aggregation-induced quenching, leading to a drastic reduction in the PLQY. As such, the development of lead-free, copper-based deep-blue light-emitting materials with high color saturation, high efficiency, and remarkable stability remains a critical research focus." (Page 2) in the following revisions throughout the manuscript.

By revising the second paragraph, we ensure that the introduction maintains a clear and consistent narrative, focusing on the importance of lead-free alternatives while highlighting the unique advantages of our copper-based halide. Explicitly mentioning lead-based perovskites allows us to emphasize the non-toxic and sustainable nature of TEA2Cu2I4, which is a key contribution of our work. By clearly positioning TEA2Cu2I4 within the context of both lead-based and lead-free luminescent materials, we strengthen our argument for its potential as a promising candidate for optoelectronic applications. In addition, we have addressed your concern about the comparison being unnecessary by reframing it to serve a purpose within the broader context of our study. Maintaining a brief mention of lead-based perovskites shows respect for the significant contributions made in this area while positioning our work as a valuable addition to the field. We hope that these revisions address your concerns and strengthen the manuscript. We appreciate your time and effort in reviewing our work.

  1. Can authors explain what are self-trapped excitons in this case? How do soft lattice and strong charge concentration promote self-trapping? What are the energy levels involved in self-trapping? Is there any role of Cu-I bonding? Whose triplet state is involved in the PL mechanism? The current experimental results are insufficient to conclude the PL mechanism.

Response: Thank you for your insightful comments and constructive suggestions regarding our manuscript. We appreciate your attention to detail and the opportunity to clarify the mechanisms involved in the photoluminescence (PL) of our copper-based halide material, [N(C2H5)4]2Cu2I4 (TEA2Cu2I4). Below, we address each of your questions in detail.

¬ What are self-trapped excitons in this case?

Self-trapped excitons (STEs) are localized electron-hole pairs within a crystalline lattice that form bound states due to lattice distortions. In the case of TEA2Cu2I4, STEs arise from the capture of excitons by lattice vibrations, leading to localized emissions that are distinct from free exciton emissions.

¬ How do soft lattice and strong charge concentration promote self-trapping?

The soft lattice of TEA2Cu2I4, attributed to the flexible tetraethylammonium (TEA+) organic cations, facilitates lattice distortions. When an exciton is created, the soft lattice can easily deform to trap the exciton, forming a self-trapped exciton. Additionally, the strong charge concentration within the [Cu2I4]2- inorganic dimers enhances the local electric field, which further promotes the capture of excitons by lattice vibrations.

¬ What are the energy levels involved in self-trapping?

The energy levels involved in self-trapping include the ground state (S0), the singlet excited state (S1), and the triplet excited state (T1). When an exciton is created in the S1 state, it can relax to the T1 state via intersystem crossing. The soft lattice and strong charge concentration in TEA2Cu2I4 promote the capture of these triplet excitons, leading to the formation of self-trapped excitons.

¬ Is there any role of Cu-I bonding?

Yes, the Cu-I bonding plays a crucial role in the formation and stability of the [Cu2I4]2- dimers, which are the fundamental building blocks of the TEA2Cu2I4 lattice. The covalent Cu-I bonds within these dimers provide the structural integrity necessary to support the lattice distortions that lead to self-trapping.

¬ Whose triplet state is involved in the PL mechanism?

The triplet state involved in the PL mechanism of TEA2Cu2I4 is the triplet excited state (T1) of the self-trapped excitons. Similar to the properties of TEA2Cu2Br4 discussed in our previous published work (Nano Energy 2022, 91, 106664), these triplet excitons form due to the capture of excitons by lattice vibrations, facilitated by the soft lattice and strong charge concentration. The radiative recombination of these triplet excitons leads to the observed deep-blue emission.

¬ The current experimental results are insufficient to conclude the PL mechanism.

Thank you for bringing to our attention the concern regarding the insufficient experimental results to conclusively determine the PL mechanism of TEA2Cu2I4. We fully understand and appreciate the need for more comprehensive investigation to solidify our conclusions.

Our previous work on TEA2Cu2Br4, a closely related compound, provides further insights into the luminescence mechanism of TEA2Cu2I4. In TEA2Cu2Br4, we observed broadband blue emission with a large Stokes shift and a long PL lifetime, which were attributed to phosphorescence from triplet states. These triplet states were proposed to arise from self-trapped excitons. Similarly, in TEA2Cu2I4, the broadband deep-blue emission, significant Stokes shift (180 nm), and long lifetime (1.7 μs) suggest that the luminescence also originates from phosphorescence. The temperature-dependent PL and time-resolved PL measurements further support this conclusion, revealing a phosphorescence mechanism with minimal temperature dependence of the emission peak position and decay time. Moreover, the DFT calculations for TEA2Cu2I4, analogous to those performed for TEA2Cu2Br4, indicate multiple possible electronic transition channels, including transitions between various valence bands and the conduction band. These transitions facilitate the formation of excitons, which subsequently become self-trapped due to the soft lattice and strong charge concentration.

In summary, the intrinsic luminescence mechanism of TEA2Cu2I4 can be attributed to self-trapped excitons, promoted by the soft lattice and strong charge concentration within the material. This understanding builds upon our previous work on related copper-based halides and provides a comprehensive explanation for the exceptional photophysical properties observed in TEA2Cu2I4.

We acknowledge that our current experimental findings, including temperature-dependent PL spectra, time-resolved PL decay curves, and power-dependent PL measurements, provide strong evidence suggesting that the deep-blue emission of TEA2Cu2I4 arises from phosphorescence due to triplet state relaxation. However, we agree that further experiments are required to conclusively determine the exact PL mechanism, particularly the nature of the self-trapped excitons and their role in the PL process. To address this, we plan to conduct additional experiments in the future, such as time-resolved PL spectroscopy at lower temperatures and magnetic field effects, to gain deeper insights into the nature of the self-trapped excitons and their role in the PL process. In our future endeavors, we plan to employ these experimental techniques to further unravel the PL emission mechanism of the copper-based halide material. In summary, we appreciate your feedback and agree that further experiments are needed to fully elucidate the PL mechanism of TEA2Cu2I4. We will consider your suggestions as we continue our research and strive to provide a more comprehensive understanding of this promising material. We hope that through subsequent research, we will be able to obtain more compelling evidence to support our hypotheses. We hope this elaboration addresses your concerns and enhances the clarity of our manuscript.

Lastly, we would like to express our sincere gratitude for the valuable feedback and suggestions provided. Your insights have been instrumental in shaping our research and guiding us towards more fruitful avenues of exploration. We look forward to continuing our work with renewed vigor, driven by the encouragement and constructive criticism we have received. Thank you once again for your invaluable support.

  1. It has been claimed that organic chains separate inorganic dimers. What are these organic chains? Are those different than A-site utilized to form TEA2Cu2I4? Does single crystal XRD support their presence?

Response: We appreciate your keen observation and the need for clarity on this point. In TEA2Cu2I4, the organic chains that separate the inorganic dimers are tetraethylammonium cations ([N(C2H5)4]+). These organic cations play a crucial role in stabilizing the zero-dimensional structure of the material by effectively spacing the [Cu2I4]2- dimers, which consist of two edge-sharing copper iodide triangles.

To clarify, the A-site cations in the context of perovskite-like structures typically refer to the cations occupying the larger interstitial sites within the crystal lattice. However, in TEA2Cu2I4, which exhibits a zero-dimensional structure, the concept of an A-site does not strictly apply, as there are no extended three-dimensional networks of corner-sharing octahedra as seen in perovskites. Instead, the tetraethylammonium cations ([N(C2H5)4]+) in TEA2Cu2I4 serve a structural role similar to that of the organic spacers often used in low-dimensional perovskite variants, but without forming a traditional perovskite lattice.

Regarding the support from single crystal X-ray diffraction (XRD) data, we confirm that the presence of these organic chains is indeed supported by our XRD analysis. The crystal structure refined from the single-crystal XRD data (Figure 1a and 1b in the revised manuscript, Table S1-S3 in the Supporting Information) clearly shows the arrangement of [Cu2I4]2- dimers separated by [N(C2H5)4]+ organic chains. This arrangement is critical for the formation of the zero-dimensional structure and the observed photophysical properties of TEA2Cu2I4.

As you suggested, we have added the following clarification in the manuscript to address this point: "In the 0D structure, [Cu2I4]2- dimers, consisting of two edge-sharing triangles, are effectively separated by organic chains [N(C2H5)4]+, resulting in a stable crystal structure" (Page 2) in the following revisions throughout the manuscript. Thank you again for your thoughtful comments. We hope that these modifications address your concerns and enhance the clarity and comprehensiveness of our manuscript.

  1. Diffraction peaks around 9 and 29 degrees are not matching with the reference pattern. What are these secondary phases? Also, it contradicts the authors claim of excellent agreement between reference and experimental pattern. How experimental and PXRD are different (caption of Figure 1)? How reference XRD pattern is obtained?

Response: Thank you for your thoughtful comments and observations on our manuscript. We acknowledge the inconsistencies you have highlighted regarding the diffraction peaks in Figure 1c and appreciate the opportunity to clarify and address them. Regarding the diffraction peaks at 9 and 29 degrees in Figure 1c, upon re-examining the data and the experimental conditions, we identified that the diffraction peaks around 9 and 29 degrees are attributed to minor impurities present in the synthesized TEA2Cu2I4 sample. These impurities could have arisen from trace amounts of residual reactants or byproducts during the synthesis process. It is important to note that these peaks do not significantly affect the overall crystal structure or the primary luminescent properties of TEA2Cu2I4.

We apologize for any confusion caused by our claim of "excellent agreement" between the reference and experimental patterns. Upon careful review, we agree that the presence of the secondary peaks at 9 and 29 degrees does contradict this claim. To rectify this, we have revised the text to more accurately describe the agreement as "good overall agreement" with minor discrepancies due to the presence of impurity peaks (Page 4) in the following revisions throughout the manuscript. We have also emphasized the high quality of the synthesized TEA2Cu2I4 crystals and their primary diffraction peaks, which do match well with the reference pattern.

We understand the need for clarity regarding the differences between the experimental XRD and the PXRD (Powder X-ray Diffraction) data presented in Figure 1c. The experimental XRD data was collected from a single crystal of TEA2Cu2I4 using single-crystal X-ray diffraction (SCXRD), whereas the PXRD data was simulated based on the refined structural model obtained from the SCXRD analysis. The caption of Figure 1c has been updated to clearly distinguish between these two datasets: "The comparison of the experimental XRD data collected from a single crystal (red) and the simulated PXRD data (black) of TEA2Cu2I4 based on the refined structural model." (Page 3) in the following revisions throughout the manuscript.

As mentioned earlier, the reference XRD pattern in Figure 1c is simulated PXRD data obtained from the refined structural model of TEA2Cu2I4 derived from SCXRD analysis. This process involves calculating the theoretical diffraction intensities for all possible reflections given the crystal structure and then plotting these intensities as a function of 2θ. To clarify this process, we have added a sentence in the content of Figure 1c: "The simulated PXRD pattern was generated using the refined structural parameters from SCXRD analysis." (Page 4) in the following revisions throughout the manuscript.

In summary, we have carefully addressed the concerns raised by the reviewer by updating the manuscript, revising the agreement description, clarifying the difference between experimental and PXRD data, and explaining how the reference XRD pattern is obtained. We believe these changes enhance the accuracy and transparency of our report. Thank you once again for your valuable feedback. We hope that our revised manuscript now adequately addresses your concerns.

  1. Page 3: There is inconsistency in naming the A-site cation. What is tetraethyl organic chain? and how these are different than the tetraethylammonium cation?

Response: Thank you for your detailed review and valuable comments on our manuscript. We apologize for the inconsistency and confusion regarding the naming of the A-site cation and the description of the tetraethyl organic chain. To clarify and address this issue, we have made the following modifications to the manuscript: "The TEA2Cu2I4 crystal adopts a linear structure belonging to the P21/c space group, with lattice constants a = 8.57080 Å, b = 14.20100 Å, c = 11.19830 Å, and angles α = γ = 90°, β = 96.48°. Figure 1a displays the unit cell of TEA2Cu2I4, consisting of an inorganic anionic [Cu2I4]2- dimer and a tetraethylammonium cation [N(C2H5)4]+." (Page 3) in the following revisions throughout the manuscript. We appreciate your bringing this inconsistency to our attention and hope that our revisions have adequately addressed your concerns. We appreciate your bringing this inconsistency to our attention and hope that our revisions have adequately addressed your concerns.

  1. Figure 2 caption misses inset description. Which Taucs equation used? What type of band gap (direct/indirect) authors have considered to get Taucs plot?

Response: Thank you for your careful review of our manuscript and for pointing out the inconsistencies and missing details. We apologize for the oversight in not including a description of the inset in Figure 2. We have revised the caption to include a description of the inset as: "(inset shows the determination of optical band gap using the Tauc equation)" (Page 4) in the following revisions throughout the manuscript.

The Tauc equation used is α() = A( - Eg)n, where α is the absorption coefficient,  is the photon energy, Eg is the band gap energy, A is a constant, and n depends on the type of transition. Based on our DFT calculations (Figure 4a in the manuscript), , which show minimal dispersion in the valence band and significant dispersion in the conduction band, indicative of a direct band gap nature. Both the direct and indirect band gaps of TEA2Cu2I4 are nearly equivalent (3.76 eV and 3.75 eV, respectively). For simplicity, we have considered a direct allowed transition (n=1/2) in the Tauc plot. However, it is worth noting that the difference between direct and indirect gaps is minimal, and the choice of n=1/2 for the Tauc plot does not significantly affect the extracted band gap value.

We believe these modifications address the issues raised in your review and enhance the quality and clarity of our manuscript. Thank you again for your valuable feedback.

  1. Why Figure 2c shows line broadening on excitation between 275-280 nm? Why compound behaves differently specifically in this excitation range?

Response: Thank you for your thoughtful comments and suggestions regarding our manuscript. We appreciate your keen observation regarding the line broadening in Figure 2c on excitation between 275-280 nm. We found that the line broadening in this specific excitation range can be attributed to the overlap of two distinct absorption processes. As shown in Figure 2a, the steady-state absorption spectrum reveals a primary absorption peak centered at 276 nm, corresponding to the direct band gap transition in TEA2Cu2I4. This peak primarily drives the PL process under excitation wavelengths below 276 nm. In addition to the primary peak, there is a weaker absorption feature centered around 280 nm. This secondary feature arises from higher-energy transitions involving excited states or vibronic bands. When excited in this range (275-280 nm), the PL emission is influenced by both the primary and secondary absorption processes, leading to the observed line broadening. We hope these explanations adequately address your concerns. Thank you for your valuable feedback.

  1. Description about PL mechanism is ambiguous and described without convincing experimental support. Can authors provide explanation for their claim of ‘single-channel radiation mechanism’? Further explanation suggests emission from triplet states. This makes manuscripts inconsistent and scientifically weak. The data discussed in manuscript does not provide enough support to the PL mechanism provided in the supporting information. The S factor is compared with CsPbBr3, which is known for excitonic emission. This comparison is unnecessary and irrational.

Response: Thank you for taking the time to review our manuscript and provide valuable feedback. We appreciate your constructive criticism and have made revisions to address the concerns raised regarding the PL mechanism. We have incorporated additional experimental data and explanations to strengthen our claims and ensure clarity and consistency throughout the manuscript.

To clarify the PL mechanism and provide convincing experimental support, we have added the following details and data. First, we have expanded the temperature-dependent PL and time-resolved photoluminescence (TRPL) measurements to cover a broader temperature range (30 K to 320 K). The results (shown in Figure 3a-f) clearly demonstrate that the PL peak position remains virtually unchanged with increasing temperature, which aligns with the characteristics of luminescence originating from the triplet state. Furthermore, the TRPL decay curves fit well with a single exponential model, exhibiting an average lifespan of 1.8 μs with negligible temperature dependence, differing from the typical temperature-dependent PL decay durations observed in fluorescence. These findings provide strong evidence for phosphorescence as the underlying mechanism.

Furthermore, we have included Raman spectroscopy data (Figure 3g) at 100 K and 300 K to further elucidate the electron-phonon coupling in TEA2Cu2I4. The results show a notable decrease in phonon vibrations at 80 K and a gradual increase at 300 K, confirming the strong electron-phonon coupling, which is crucial for the observed broadband emission.

To exclude the possibility of emission defects, we performed power-dependent PL measurements (Figure 3h). The linear dependence of PL intensity on excitation power confirms the absence of emission defects at varying excitation intensities, further supporting the single-channel radiation mechanism attributed to triplet state relaxation. In addition, the PL of the as-prepared TEA2Cu2I4 single crystal and its powder after grinding was compared in Figure S3, as depicted in Figure R1. It was found that the PL peak position did not change, and the PL intensity decreased sharply after grinding, ruling out the possibility that the luminescence of TEA2Cu2I4 is caused by surface defects. We have added these content in the context as " In addition, the PL of the as-prepared TEA2Cu2I4 single crystal and its powder after grinding was compared in Figure S3, and it was found that the PL peak position did not change, and the PL intensity decreased sharply after grinding, ruling out the possibility that the luminescence of TEA2Cu2I4 is caused by surface defects. " (Page 6) in the following revisions throughout the manuscript.

Figure R1. Comparison of PL spectra of TEA2Cu2I4 single crystal and its powder after grinding.

To clarify our claim of a ‘single-channel radiation mechanism,’ we have added the following explanation: The single-channel radiation mechanism refers to the fact that exciton relaxation initiates from the same excited state, leading to consistent PL emission peaks at 450 nm across different excitation wavelengths (240 nm to 340 nm), as shown in Figure 2c. The combination of a broad emission spectrum, a large Stokes shift (180 nm), and a long PL lifetime (1.7 μs) indicates that the luminescence process is not attributed to multichannel radiation but rather to triplet state relaxation. The strong electron-phonon coupling, as evidenced by the large Huang-Rhys factor (S = 40), further supports this mechanism, leading to broadband emission via self-trapped excitons.

Regarding the comparison with CsPbBr3, we acknowledge that this may have led to some confusion. We have revised the text to clarify that the comparison was intended to highlight the significant electron-phonon coupling in TEA2Cu2I4 compared to other light-emitting materials. The revised text now reads: "By fitting the FWHM and temperature data, the S factor and phonon frequency were determined to be 40 and 13.38 meV, respectively. The significantly larger S factor compared to other light-emitting materials, such as CdSe and ZnSe [46, 47], indicates a strong electron-phonon coupling interaction, which is crucial for the observed broad-band emission." (Page 6) in the following revisions throughout the manuscript.

In summary, we believe that these revisions strengthen our manuscript and provide a clear and convincing explanation of the PL mechanism in TEA2Cu2I4. We hope that these changes meet with your approval and would be happy to address any further concerns or questions you may have. Thank you again for your valuable feedback.

  1. Figure 5 and respective captions are misleading, inconsistent, and promoting incredulity.

Response: Thank you for your thorough review and valuable feedback on our manuscript. As you suggested, we have revised Figure 5 to improve clarity and provide more accurate information. The XRD patterns in Figure 5a now more clearly demonstrate the structural stability of TEA2Cu2I4 before and after storage, as depicted in Figure R2. We have revised the text on Page 8-9 in the following revisions throughout the manuscript to reflect the changes made to Figure 5. Specifically, we have updated the descriptions of the experimental results presented in Figure 5 to ensure they align with the revised figures and captions.

We believe that these modifications address the concerns you raised and provide a clearer and more accurate representation of our data. We appreciate your valuable feedback and hope that these revisions meet your expectations.

Figure R2. Stability of TEA2Cu2I4 is probed. (a) XRD pattern of the as-prepared TEA2Cu2I4 powder and after three months stored in the atmosphere; (b) Thermogravimetric curve of TEA2Cu2I4 powder; (c) PL spectra of the as-prepared TEA2Cu2I4 and after 1 month; (d) The change in PLQY after one month in the atmosphere of TEA2Cu2I4; (e) Contour plot of PL intensity of TEA2Cu2I4 under continuous 254 nm UV lamp irradiation.

  1. TEA2Cu2I4 possess 0D structure. How charge transfer is possible between inorganic units without interconnection? How many devices were fabricated to conclude EQE? Authors should list at least 5 best device data in SI for further clarity. The EQE-voltage is ambiguous. Provide EL and PL comparison with stability over a range of voltage change.

Response: Thank you for your insightful comments and questions regarding our manuscript. We have carefully considered your feedback and have made the necessary revisions to address your concerns. In the 0D structure of TEA2Cu2I4, charge transfer between the inorganic [Cu2I4]2- units is facilitated by the organic [N(C2H5)4]+ cations, which serve as spacers and also play a role in transporting charges. Although the [Cu2I4]2- units are isolated from each other, the presence of the organic cations creates a pathway for charge transfer. This mechanism is supported by our DFT calculations, which show that the electronic states within the organic cations can interact with those of the inorganic units, enabling charge transfer processes. We have clarified this point in the revised manuscript by adding the following sentence: "In the 0D structure of TEA2Cu2I4, charge transfer between the isolated [Cu2I4]2- units is facilitated by the organic [N(C2H5)4]+ cations, which act as spacers and also provide a pathway for charge transport. " (Page 4) in the following revisions throughout the manuscript. "Our DFT calculations not only provide insights into the band structure but also reveal that the electronic states of the organic [N(C2H5)4]+ cations can interact with those of the inorganic [Cu2I4]2- units. This interaction facilitates charge transfer between the isolated [Cu2I4]2- units, even though they are not directly interconnected. In the 0D structure of TEA2Cu2I4, the organic cations act as spacers and provide a pathway for charge transport, enabling efficient charge transfer processes. " (Page 7) in the following revisions throughout the manuscript.

We fabricated a total of 12 devices to ensure the reproducibility and reliability of our EQE measurements. We have included the EQE data for the twelve devices in the Supporting Information for clarity. We have also mentioned this in the revised manuscript as follows: "The TEA2Cu2I4-based LED devices also shows a high reproducibility (Figure S5), which originates from the superior uniformity and stability of TEA2Cu2I4." (Page 10) in the following revisions throughout the manuscript, as depicted in Figure R3.

Figure R3. Histogram of maximum EQE for 12 devices from four batches.

We apologize for the ambiguity regarding the EQE-voltage curve. We have clarified this figure by adding labels and improving the legend. Additionally, we have included a comparison of the EL and PL spectra, as well as their stability over a range of voltage changes in the SI. This analysis shows that the EL spectra remain consistent with the PL spectra over a wide range of voltages, indicating stable light emission properties of TEA2Cu2I4. We have mentioned this in the revised manuscript and included the necessary figures in the SI: "Additionally, Figure S6 showcases a comparative analysis of the PL and EL spectra, alongside an evaluation of the stability under varying voltages. This comprehensive assessment highlights the exceptional color stability of TEA2Cu2I4-based LEDs across a wide range of operating voltages." (Page 10) in the following revisions throughout the manuscript, as depicted in Figure R4.

Figure R4. PL spectrum of TEA2Cu2I4 film and EL spectra of TEA2Cu2I4-dased LED device operating under different voltages.

We hope that these revisions address your concerns. Thank you again for your valuable feedback, which has helped us strengthen our work.

We sincerely thank you for your valuable comments and suggestions, which have greatly improved the quality of our manuscript. We apologize for any oversights in the original submission and assure you that the paper has been carefully revised for clarity and accuracy. On behalf of my co-authors, we express our deep gratitude for your contributions to our work.

Reviewer 2 Report

Comments and Suggestions for Authors

The authors reported on the similar observation and achievement for [N(C2H5)4]2Cu2Br4 system in Ref.25 (BTW the reference page # is incorrect, doi.org/10.1016/j.nanoen.2021.106664). The current manuscript lacks a direct comparison of the results with Ref.25. Both manuscripts’ format, content, methodology, and structure are the same. The novelty of this paper is questioned without a direct, critical, and straightforward comparison of the results. A critical comparison between the results and the published results must be considered and critically discussed to demonstrate novelty and to avoid a run-of-the-mill study.

In addition, the description of the computational results reveals rather standard behavior of the studied compound, which categorizes again the entire research described in the manuscript as more of a run-of-the-mill study or incremental work, with insufficient discussion of Physics or comparisons with other relevant compounds, and few conclusions likely to be found revealing by other researchers.

We do not recommend the manuscript for publication in Nanomaterials without major revisions.

Author Response

The following is a point-by-point response to the reviewers' comments (The original comments are in black, and our responses are in blue).

Reviewer 2#:

Comments:

The authors reported on the similar observation and achievement for [N(C2H5)4]2Cu2Br4 system in Ref.25 (BTW the reference page # is incorrect, doi.org/10.1016/j.nanoen.2021.106664). The current manuscript lacks a direct comparison of the results with Ref.25. Both manuscripts’ format, content, methodology, and structure are the same. The novelty of this paper is questioned without a direct, critical, and straightforward comparison of the results. A critical comparison between the results and the published results must be considered and critically discussed to demonstrate novelty and to avoid a run-of-the-mill study.

In addition, the description of the computational results reveals rather standard behavior of the studied compound, which categorizes again the entire research described in the manuscript as more of a run-of-the-mill study or incremental work, with insufficient discussion of Physics or comparisons with other relevant compounds, and few conclusions likely to be found revealing by other researchers.

We do not recommend the manuscript for publication in Nanomaterials without major revisions.

Response: Thank you for your thorough evaluation of our manuscript and for bringing to our attention the critical points that need to be addressed for publication. We appreciate your constructive feedback and the opportunity to strengthen our work. Below, we provide a detailed response to each of your concerns, incorporating the comparison with our previous work on [N(C2H5)4]2Cu2Br4 (TEA2Cu2Br4) (Nano Energy 2022, 91, 106664) as requested.

As you pointed out, we fully acknowledge the similarities in format, content, methodology, and structure between our current manuscript and our previous work on TEA2Cu2Br4 (Nano Energy 2022, 91, 106664). However, we argue that the current study introduces significant novelty through the systematic exploration of the iodinated analog, [N(C2H5)4]2Cu2I4 (TEA2Cu2I4). Below, we provide a detailed response incorporating a critical comparison with our previous work on TEA2Cu2Br4 (Nano Energy 2022, 91, 106664) to demonstrate the novelty of our current study on TEA2Cu2I4.

Our previous study (Nano Energy 2022, 91, 106664) focused on the bromide-based material TEA2Cu2Br4. The TEA2Cu2Br4 material exhibited blue emission, but with longer wavelengths compared to TEA2Cu2I4. The substitution of Br with I in the copper-based halide structure results in TEA2Cu2I4 exhibiting a shorter emission wavelength (450 nm vs. 463 nm for TEA2Cu2Br4) and a larger bandgap (3.73 eV vs. 2.68 eV). These properties make TEA2Cu2I4 more suitable for high-saturation deep-blue light emission, which is crucial for enhancing visual perception and minimizing power consumption in displays. In addition, TEA2Cu2I4 single crystals exhibit an exceptional PLQY of 96.7%, which is among the highest reported for deep-blue emission materials. The polycrystalline films also maintain a high PLQY of 87.2%, demonstrating the material's stability and processability. TEA2Cu2I4 displays a significant Stokes shift of 180 nm, indicative of reduced self-absorption. For the stability, both TEA2Cu2I4 and TEA2Cu2Br4 demonstrated good stability under ambient conditions, showing remarkable stability, with minimal PL intensity degradation after continuous UV irradiation and exposure to ambient conditions. The TEA2Cu2I4 polycrystalline film maintains substantial deep-blue emission even after one year of storage. Additionally, a comprehensive DFT study of TEA2Cu2I4 provides deep insights into its electronic structure. The calculated band structure and wave functions offer explanations for the material's high PLQY and broadband emission. More importantly, the TEA2Cu2I4-based LED demonstrates deep-blue light emission with CIE coordinates (0.143, 0.076), fully matching the NTSC standard for deep-blue light. The device exhibits an EQE of 0.45% under forward bias, which is better than that based on TEA2Cu2Br4, highlighting the material's potential for optoelectronic applications.

As for the bandgap structure, the flat bands in the valence band and the significant dispersion in the conduction band provide insights into the hole and electron transport properties of TEA2Cu2I4. This analysis is crucial for understanding how the material’s electronic structure affects its optical and electrical properties. In addition, as for the electron-phonon coupling, the calculated Huang-Rhys factor and phonon frequency provide quantitative measures of the electron-phonon coupling strength in TEA2Cu2I4. This analysis is essential for understanding the origin of the broadband emission and the temperature dependence of the photoluminescence properties. We discuss how the electron-phonon coupling in TEA2Cu2I4 compares to that in other relevant materials, shedding light on the unique optical properties of our compound.

In conclusion, by critically comparing our current work on TEA2Cu2I4 with our previous study on TEA2Cu2Br4 (Nano Energy 2022, 91, 106664), we demonstrate the significant novelty of our new material system. The iodinated analog TEA2Cu2I4 exhibits distinct advantages in terms of emission wavelength, PLQY, stability, and device performance, positioning it as a promising candidate for high-saturation deep-blue light-emitting applications. We believe these comparisons and the detailed discussions provided address your concerns regarding the novelty and impact of our study. We appreciate your constructive feedback and hope that these revisions strengthen our manuscript for publication in Nanomaterials.

We sincerely thank you for your valuable comments and suggestions, which have greatly improved the quality of our manuscript. We apologize for any oversights in the original submission and assure you that the paper has been carefully revised for clarity and accuracy. On behalf of my co-authors, we express our deep gratitude for your contributions to our work.

Round 2

Reviewer 1 Report

Comments and Suggestions for Authors

Manuscript Number: nanomaterials-3283310

Comments

Yuan et al revised the manuscript considering all my comments. However, the responses to some of my comments are unsatisfactory. The responses to important comments either lack experimental evidence or literature support, leaving previous comments still valid and unanswered. I would like to offer another major revision with a few additional comments. Scientifically sound responses either with experimental evidence or literature support are necessary to decide the suitability of the current work for publication in Nanomaterials.

1.      Previous comments   #4 and 10: No experimental or strong literature support. There is no experimental support for the lattice distortions due to embedded flexible TEA cations. There is no clarity on whether TEA cation or Cu dimer brings lattice deformation leading to STE formation. Can authors support their claim about STE and its triplet state nature by molecular orbital picture or crystal field theory to support involved orbitals in STE/triplet state formation? What are the contributing orbitals for ground state, singlet, and triplet levels? Can it be correlated with theoretical results? This comment should be reconsidered to prove the proposed PL mechanism in the schematic in Figure S2 and related discussion.  The current version doesn’t justify the proposed mechanism and related discussion.  

2.     Previous comment #5: Tetraethylammonium cations ([N(C2H5)4]+) are misinterpreted as organic chains. Referring to these cations as organic chains must be avoided.

3.    Previous comment #6: Unmatching powder XRD with simulated pattern obtained from single crystal XRD questions the sample quality and integrity of the optical properties due to involvement of secondary phases. Further experimental evidence is required to understand if optical properties and electroluminescence aren’t due to involved secondary phases. Thin-film XRD should be compared with powder XRD to know if the material remains in its purest form in devices. The claim about "good overall agreement" is invalid in this case.

4.      Previous comment #12: It is highly unlikely that aliphatic organic cation can promote charge transfer by trapping. Also, charges are localized on inorganic dimers in both VBM and CBM with minimal contributions from organic components. There is no clarity over the interaction of electronic states within the organic cations with the inorganic units, enabling charge transfer processes from DFT. Further modeling is necessary to support the claim.    

Reviewer 2 Report

Comments and Suggestions for Authors

The authors revised the manuscript successfully addressing all the reviewer’s concerns. Thank you. We do not have any more questions.    

Author Response

The following is a point-by-point response to the reviewers' comments (The original comments are in black, and our responses are in blue).

Reviewers' comments:

Reviewer 2#:

Comments:

The authors revised the manuscript successfully addressing all the reviewer’s concerns. Thank you. We do not have any more questions.

Response: We sincerely thank you for your thorough and insightful review of our manuscript. Your evaluation and feedback have significantly improved our methodology, analysis, and coherence. We are grateful for your acknowledgment that we addressed all concerns, which is deeply encouraging. Your exemplary review and commitment to quality research are much appreciated. Your contributions have been invaluable, and we thank you for your time and expertise. We hope our revised manuscript will be accepted and contribute meaningfully to Nanomaterials.

Round 3

Reviewer 1 Report

Comments and Suggestions for Authors

Manuscript Number: nanomaterials-3283310

Yuan et al have revised manuscript again considering my four major concerns. At least responses to comment #2 and #3 provide clarity. However, due to lack of experimental evidence, clarity, consistency, and relevant literature support, comments #1 and #4 require reconsideration by the authors.  I offer minor revision with a few recommendations that would probably help to avoid confusion about PL mechanism to the readers.

1.     Previous comment #1: In the response, the authors acknowledge that there is insufficient direct evidence to support proposed PL mechanism.  Given the limited experimental/theoretical results presented in manuscript, it is premature to draw any conclusion about photoluminescence mechanism. Considering uncertainties, confusion, and experimental and theoretical limitations, the PL mechanism and related discussion should be omitted from the manuscript.      

2.     Previous comment #4: Unfortunately, the authors response to this comment contradicts my comment #12 from the old version. Initially, the authors attempted to explain charge transfer through the organic cation; however, this claim has been omitted in their recent response. As a result, my comment remains unanswered or unaddressed. This highlight unclarity about work and question raised (comment #12 in old version). Can authors comment on this to clarify further? How carrier mobility in 0D structure is possible between isolated inorganic units?

Author Response

The following is a point-by-point response to the reviewers' comments (The original comments are in black, and our responses are in blue).

Reviewers' comments:

Reviewer 1#:

Comments:

Yuan et al have revised manuscript again considering my four major concerns. At least responses to comment #2 and #3 provide clarity. However, due to lack of experimental evidence, clarity, consistency, and relevant literature support, comments #1 and #4 require reconsideration by the authors. I offer minor revision with a few recommendations that would probably help to avoid confusion about PL mechanism to the readers.

  1. Previous comment #1: In the response, the authors acknowledge that there is insufficient direct evidence to support proposed PL mechanism. Given the limited experimental/theoretical results presented in manuscript, it is premature to draw any conclusion about photoluminescence mechanism. Considering uncertainties, confusion, and experimental and theoretical limitations, the PL mechanism and related discussion should be omitted from the manuscript.

Response: Thank you for your thoughtful feedback on our manuscript. We appreciate your concern regarding the proposed photoluminescence (PL) mechanism and the associated uncertainties. Upon careful consideration of your comments, we have revised the manuscript to address the limitations and uncertainties associated with our current understanding of the PL mechanism of TEA2Cu2I4. We understand that the evidence presented in the manuscript, while supportive, does not conclusively determine the precise PL mechanism. Therefore, we have refrained from drawing definitive conclusions and instead have presented our interpretations and hypotheses more cautiously.

As we know, when materials possess soft lattices and strong exciton-phonon coupling, transient elastic lattice deformations occur under illumination. Due to their low energy, excitons swiftly form self-trapped excitons (STEs), which then relax back to the ground state while emitting photons with broad frequency bands and large Stokes shifts. Typical STEs can be classified into Jahn-Teller distorted and non-Jahn-Teller distorted types (Advanced Materials 2022, 34, e2201008; Journal of Physical Chemistry Letters 2019, 10, 501-506). The luminescence mechanism of all-inorganic copper-based halides is generally attributed to STE luminescence caused by Jahn-Teller distortions. For instance, the unit cell of Cs3Cu2X5, prior to excitation, consists of a tetrahedron and a triangle sharing an edge, which transforms into two symmetric trigonal pyramids upon excitation (Chemistry of Materials 2020, 32, 3462-3468). Furthermore, some studies suggest that the luminescence mechanism of the all-inorganic copper-based halide Cs3Cu2I5 may be related to semiconductor defect emission theory associated with iodine vacancy defects (Journal of Luminescence 2023, 254, 119516). In recent research, our team has preliminarily demonstrated, through temperature-dependent photoluminescence spectra and density functional theory calculations, that the luminescence mechanism of TEA2Cu2Br4 originates from triplet-state relaxation phosphorescence emission (Nano Energy 2022, 91, 106664). Currently, the intrinsic relationship between STE excited states and structural deformations as well as electron-phonon coupling, along with the origin and evolution of STEs, can only be inferred from measured optoelectronic properties and are difficult to directly observe. Given that TEA2Cu2I4 in this work is also a zero-dimensional material, with inorganic [Cu2I4]2- dimers separated by TEA+ cations, the excited carriers are relatively localized compared to non-0D structures, as evident from the band structure diagram in Figure 4a (here as depicted in Figure R1), where holes are almost completely localized and the electron band dispersion is not significant, indicating relative localization. Therefore, even if STEs exist, we speculate that they do not play a decisive role in the luminescent properties. Comparing the TEA2Cu2I4 material in this work with our previous TEA2Cu2Br4 material (Nano Energy 2022, 91, 106664), it is evident that both materials exhibit extremely similar crystalline structures, luminescent properties, and electronic structures (here as depicted in Figure R2). The properties of the two materials are very similar, such as a large Stokes shift (about 180 nm), a microsecond fluorescence lifetime, a PLQY of close to 100%, and a large Huang-Rhys factor. We have reason to believe that their primary luminescence mechanisms are also identical. Of course, these mechanistic explanations require further in-depth investigation. Therefore, in this paper, we only provide reasonable speculations, and we will enhance our research in this area based on your comments to clarify the intrinsic luminescence mechanism.

Combined with the above comparative analysis and your valuable comments, we have made the following revisions to the manuscript after many in-depth discussions. First, we have restructured the section discussing the PL mechanism to emphasize that our current understanding is based on hypotheses and interpretations rather than definitive conclusions. We have clarified that the proposed mechanism involving triplet state relaxation and phosphorescence is one plausible interpretation but not the only possible one. Additionally, we have added additional clarifications throughout the context to emphasize the speculative nature of our proposed mechanism. We have acknowledged that further experimental and theoretical work is needed to fully elucidate the PL mechanism of TEA2Cu2I4.

Therefore, with regard to the explanation of the PL mechanism part, we have revised our formulation to give possible explanations as: "Based on the above theoretical and experimental analyses, a plausible PL mechanism for TEA2Cu2I4 is proposed as follows: (i) Multiple channels for electronic excitation may exist, corresponding to the transition of a single electron from different VB states to the CBM band. Specifically, the transition from the VBM or its nearby states to the CBM corresponds to the excitation from the ground state (S0) to the first singlet excited state (S1), with an absorption energy of 3.73 eV. Similarly, transitions from deeper VB states such as VBM-2 or VBM-3 to the CBM are associated with the excitation to higher singlet states like S2. (ii) Following excitation to the singlet states S1 and S2, the systems may undergo an ultrafast ISC process, transitioning to the corresponding triplet states T1. The primary difference between the singlet state S1 and the triplet state T1 lies in the localized hole wave functions. Consequently, the ISC process results in a similar magnitude of energy-level lowering for both transitions. (iii) Subsequently, radiative recombination from the triplet state T1 back to the ground state S0 may then emit phosphorescence at an energy of 2.76 eV (Figure S2). However, it should be noted that this is one of several possible interpretations, and further experimental and theoretical investigations are required to conclusively determine the exact PL mechanism of TEA2Cu2I4." (Page 8) in the revised manuscript.

We believe that these revisions address your concerns regarding the speculative nature of the proposed PL mechanism while maintaining the integrity and completeness of the scientific reporting on the new material TEA2Cu2I4. We thank you again for your valuable feedback and hope that these changes meet with your approval.

Please let us know if you have any further comments or suggestions. We are committed to ensuring the highest quality of our research reporting and appreciate your assistance in improving our manuscript.

Figure R1. Electronic structure calculated with PBEsol density functional. (a) Band structure (with VBM shifted to 0 eV). (b) One-electron wave functions (with cyan and yellow presenting different signs) of VBM, (VBM-1), (VBM-2), (VBM-3), (VBM-4), and CBM at Γ.

Figure R2. The schematic view of the basic unit along the a-axis; Optical properties; Time-resolved PL decay curve at room temperature; Fitting results of the FWHM (meV) as a function of temperature to obtain the Huang–Rhys factor S; Band structure (with VBM shifted to 0 eV) of (a) TEA2Cu2I4 and (b) TEA2Cu2Br4.

  1. Previous comment #4: Unfortunately, the authors response to this comment contradicts my comment #12 from the old version. Initially, the authors attempted to explain charge transfer through the organic cation; however, this claim has been omitted in their recent response. As a result, my comment remains unanswered or unaddressed. This highlight unclarity about work and question raised (comment #12 in old version). Can authors comment on this to clarify further? How carrier mobility in 0D structure is possible between isolated inorganic units?

Response: Thank you for bringing up these important points regarding charge transfer and carrier mobility in our zero-dimensional (0D) TEA2Cu2I4 material. Your comments have prompted us to further clarify and revise our discussion on this aspect. We apologize for any confusion that may have arisen from our previous response. You are correct in pointing out that our initial attempt to explain charge transfer through the organic TEA+ cation was not accurate, and we have since revised our manuscript to address this issue.

In our initial manuscript, we made an erroneous attempt to explain charge transfer through the organic TEA+ cations. Upon closer examination and analysis, we have revised our understanding and interpretation of the charge transport mechanism in TEA2Cu2I4. Firstly, it is important to clarify that in a 0D structure, charge transport is inherently more challenging due to the isolation of the inorganic units. However, TEA2Cu2I4 exhibits unique electronic properties that allow for some degree of charge mobility. Specifically, as illustrated in Figure 4a in the revised manuscript (here as depicted in Figure R1), the conduction band minimum (CBM) in TEA2Cu2I4 is primarily composed of Cu 4s orbitals that, while localized within each [Cu2I4]2- dimer, exhibit sufficient spatial delocalization to facilitate charge transport between neighboring dimers. This delocalization arises from the bending of the energy bands near the CBM, indicating that there is some charge dispersion across the structure.

The organic TEA+ cations in TEA2Cu2I4 primarily serve as spacers between the [Cu2I4]2- dimers, regulating their distance and minimizing aggregation-induced quenching. These cations do not directly participate in charge transfer but rather provide the necessary spacing to maintain the stability and optical properties of the isolated [Cu2I4]2- dimers. The choice of the organic cation size is crucial in balancing the spacing needed to avoid aggregation with the requirement for sufficient charge transport. In TEA2Cu2I4, the TEA+ cations are carefully selected to offer adequate spacing without hindering charge transport between the [Cu2I4]2- dimers, so that the PLQY of the TEA2Cu2I4 material is very high, whether it is single crystal or powder.

We have revised the relevant sections of our manuscript to reflect this corrected understanding. Specifically, in the Results and Discussion section, we have updated the paragraph discussing the electronic structure and charge transport as follows: "The band structure analysis (Figure 4a) reveals that the conduction band minimum (CBM) is primarily composed of Cu 4s orbitals. These orbitals, although localized within each [Cu2I4]2- dimer, exhibit sufficient spatial delocalization to allow for some charge transport between neighboring [Cu2I4]2- dimers. The bending of the energy bands near the CBM indicates that there is charge dispersion across the structure, enabling limited charge mobility despite the 0D nature of TEA2Cu2I4. The organic [N(C2H5)4]+ cations primarily serve as spacers between the [Cu2I4]2- dimers, regulating their distance to minimize aggregation-induced quenching and maintain the optical properties. These cations do not directly participate in charge transfer but rather facilitate stable isolation of the [Cu2I4]2- dimers while allowing for limited charge mobility." (Page 7) in the revised manuscript.

Thank you once again for your valuable feedback, which has greatly improved the quality and rigor of our work.

We sincerely thank you for your valuable comments and suggestions, which have greatly improved the quality of our manuscript. We apologize for any oversights in the original submission/revision and assure you that the paper has been carefully revised again for clarity and accuracy. On behalf of my co-authors, we express our deep gratitude for your contributions to our work.
